# COVID-19 News Exposure and Vaccinations: A Moderated Mediation of Digital News Literacy Behavior and Vaccine Misperceptions

**DOI:** 10.3390/ijerph20010891

**Published:** 2023-01-03

**Authors:** Yuanyuan Wu, Ozan Kuru, Dam Hee Kim, Seongcheol Kim

**Affiliations:** 1Department of Communications and New Media, Faculty of Arts & Social Sciences, National University of Singapore, Singapore 117416, Singapore; 2Department of Communication, College of Social & Behavioral Sciences, The University of Arizona, Tucson, AZ 85721, USA; 3School of Media and Communication, Korea University, Seoul 02841, Republic of Korea

**Keywords:** news literacy, COVID-19 vaccines, misinformation, digital media, news exposure

## Abstract

Being exposed to and believing in misinformation about COVID-19 vaccines is a challenge for vaccine acceptance. Yet, how countervailing factors such as news literacy could complicate “the information exposure—belief in vaccine misinformation—vaccination” path needs to be unpacked to understand the communication of scientific information about COVID-19. This study examines (1) the mediating role of belief in vaccine misinformation between COVID-19 information exposure and vaccination behavior and (2) the moderating role of news literacy behaviors. We examine these relationships by collecting data in two distinct societies: the United States and South Korea. We conducted online surveys in June and September 2021 respectively for each country (*N* = 1336 [the U.S.]; *N* = 550 [South Korea]). Our results showed a significant moderated mediation model, in which the association between digital media reliance and COVID-19 vaccination was mediated through vaccine misperceptions, and the relationship between digital media reliance and misinformed belief was further moderated by news literacy behavior. Unexpectedly, we found that individuals with stronger news literacy behavior were more susceptible to misinformation belief. This study contributes to the extant literature on the communication of COVID-19 science through probing into the mediating role of belief in vaccine-related misinformation and the moderating role of news literacy behavior in relation to COVID-19 information exposure and vaccination behaviors. It also reflects the concept of news literacy behavior and discusses how it could be further refined to exert its positive impact in correcting misinformation beliefs.

## 1. Introduction

COVID-19 pandemic have caused devastating socio-economic impacts, affecting not just human lives, but also trade, economy and businesses and other aspects [1,2]. Vaccines became the most important tool for every country to get on the path for recovery. However, the wide-spread acceptance of the vaccines cannot be taken for granted, despite the COVID-19 vaccine being considered one of the most important scientific breakthroughs and health interventions to give herd immunity and end the pandemic [3]. For example, as of early June 2021, less than half of all eligible Americans (48.6%) were “fully vaccinated” (People who have received an accepted single-dose vaccine or the second dose in a two dose COVID-19 vaccine series). Referring to the same period, other countries such as South Korea itself had low vaccinated rates (less than 10% were fully vaccinated). Even until now, there are still 32% of Americans not fully vaccinated yet [4].

The willingness to accept COVID-19 vaccines is primarily influenced by the information environment, especially given the lockdowns and remote work conditions that have made reliance on media much stronger [5]. Today, individuals increasingly rely on the internet and digital technology for health-related news and information [6], but prior research shows that social media usage could be a hurdle for vaccinations [7]. However, little is known about whether reliance on different digital media for news would influence vaccination uptake differently. In particular, health-related digital media such as health blogs and alternative health media could serve as important health information sources [8]. Recent research showed that these platforms could fuel health misinformation and affect vaccine intentions as well [9,10,11]. Therefore, this study first investigates how COVID-19 news exposure to diverse digital media, including social media, messaging applications, health blogs, and alternative health media, helps explain vaccination behaviors.

Second, we focus on beliefs in vaccine-related misinformation as an important factor in determining the relationship between news exposure and vaccination behavior. According to the health belief model (HBM), the reason why people will take action to prevent, to screen for, or to control illness conditions could be directly explained by their perceptions such as perceived barriers to health actions [12]. Misinformation is identified as one important perceived barrier to vaccinations through the lens of HBM [13]. Additionally, specific cues such as media exposure can also impact the final action an individual takes, and communication scholars tend to conceptualize external cues as the factor that could influence behavior through people’s perceptions [14]. During health crises such as the COVID-19 pandemic, accessing reliable health information becomes more challenging, whereas vaccine-related misinformation could be easily gained not only from social media but also from other digital platforms such as alternative health media [15,16,17]. Reliance on digital platforms for news acts as a barrier to vaccination to the extent that they may include misinformation about COVID-19 vaccines. Research shows that 409 English-language anti-vaccination social media accounts had 58 million followers [18]. A high level of misinformation exposure was indeed reported; for example, 73% of Americans had some exposure to misinformation about COVID-19 vaccines in the past 6 months [19]. The prevalence of vaccine misinformation makes it poignant to understand whether misinformation belief could mediate the relationship between news exposure and vaccination behavior.

Third, we examine the exposure—misinformation belief—vaccination path by taking into account a potentially countervailing factor: individuals’ news literacy behaviors. This is because COVID-19 has led to an unprecedented surge of publications across all areas of knowledge, and individuals are bombarded with a deluge of information about the pandemic over the digital media. So it is crucial to examine how skillfully people dealt with the information overload [20,21]. Researchers have pointed out news literacy could be a proactive solution to combatting misinformation [22,23], and a few recent studies explored the moderating role of literacy-related factors in the context of media exposure [15,24]. However, no study has examined the role of news literacy behaviors so far. News literacy behavior is defined as “the behaviors that occur when people engage with news content in a critical and mindful manner”, which aligns with the core outcome of news literacy programs [25]. Typical examples of news literacy behavior include using diverse news sources and verifying news content [25]. Compared to how much knowledge and skills individuals have in navigating the news information environment (i.e., news literacy) and how much they apply news literacy in practice (i.e., news literacy behavior) could be more critical in evaluating news and discerning misinformation.

In sum, we find that there is a lack of research in current literature to (1) understand the mediating role of belief in vaccine misinformation in the relationship between the use of multiple digital media, especially health-specific media, for COVID-19 information and vaccination behavior, and (2) the moderating role of news literacy behavior on this mediated path. This study offers a conceptual model that explains how exposure to different digital media is linked to vaccination via belief in misinformation. We also point out this mediated mechanism can be moderated by news literacy behavior. Further, we compare empirical evidence from two distinct societies (the U.S. vs. South Korea) to see if the model holds in different contexts.

## 2. Literature Review

In the literature review, we first explain the mediating path to clarify our important contributions, and then discuss how and what type of digital media is being studied in the next section. The last part of the literature review explores the moderating role of digital news literacy behavior

### 2.1. News Exposure and Vaccinations: The Mediating Role of Misinformation Belief

News exposure from online media might increase vaccine hesitancy as people can obtain a large amount of information that vary largely in quality [26]. Numerous studies have investigated the relationship between digital media reliance and vaccination-related outcomes (e.g., vaccine-related misinformed belief, and vaccine hesitancy, intention, or behavior) [7,27]. However, many studies only measured general media usage or reliance, rather than media reliance specifically for COVID-19 news [28]. This makes it harder to understand the extent to which COVID-19 specific information plays a role in beliefs and behaviors pertaining to the pandemic. Hence, we examine how COVID-19 news exposure to different types of digital media could influence COVID-19 vaccination behaviors.

We rely on the health belief model (HBM) to clarify how news exposure through different media could predict health behaviors such as vaccination. The health belief model (HBM) posited that an individual’s health behavior could be directly influenced by (1) perceived susceptibility and (2) perceived severity of illness conditions, as well as (3) perceived benefits and (4) perceived barriers of the health behavior [12]. Communication approaches conceptualized news use as a predictor of these four dimensions in HBM, constituting a mediation model [14]. A meta-analysis of HBM related papers revealed that “perceived barriers” were the most powerful predictor of behavior [29]. While the meaning of perceived barriers may vary in different contexts, prior studies examining COVID-19 vaccine hesitancy through the lens of HBM identified misinformation as a major perceived barrier to vaccine uptake [13]. Misinformation is defined as information that is erroneous or incorrect and not supported by the best available evidence and expert opinion [30]. False claims about COVID-19 vaccines surge on media such as linking vaccines with autism and infertility, or advocating natural immunity in replace of vaccines [31,32,33]. Empirical evidence supports that misperception of vaccines as perceived barriers contributes to low vaccine coverage [34]. High susceptibility to misinformation leads to high vaccine hesitancy and low compliance with health guidance [35,36].

Hence, misperception of vaccines, to the extent that it is fueled by media exposure, acts as an important barrier. To the best of our knowledge, this pathway has surprisingly not been studied so far. Although scholars have investigated how belief in misinformation could negatively predict vaccination intention and behavior [37], and how it can be predicted by news exposure [38], COVID-19 misinformation belief has not been examined as a mediator between news exposure and vaccination behavior. Particularly given the prevalence of vaccine-related online misinformation during the pandemic, it becomes crucial to understand the potential mediating role of belief in misinformation to (1) tease out if and to what extent beliefs act as a mediator and (2) to determine the direct and indirect effect of media exposure on vaccine behavior. In the next section, we will examine in detail how news exposure through diverse digital media could be differently associated with misinformation belief and vaccination behavior.

### 2.2. Reliance on Digital Media for COVID-19 News

Use of social media and internet-based news sources is likely to increase during crises such as the COVID-19 pandemic [39]. Next, we will demonstrate the different characteristics of these diversified digital media, and how they might impact individuals’ beliefs in health misinformation, especially vaccine-related ones.

#### 2.2.1. Reliance on Social Media and Messaging Applications

Social media is a major source of misinformation due to much freedom it lends to users and lack of gate-keeping system [40,41]. It was shown that vaccine-related misinformation was widely disseminated on social media and misinformation spreaders had a stronger community structure connecting information and users on social media compared to fact-checkers [42]. Thus, it is not surprising that social media use predicted stronger COVID-19 vaccine hesitancy [43].

Although often categorized as part of social media, messaging applications such as WhatsApp provide a more private, intimate, and controlled “safe” environment for communication, and the information from messaging applications is more likely to come from socially homogeneous sources [44]. Thus, it might be easier for users to believe in misinformation shared by their friends on a messaging application given that individuals tend to use source identity as a heuristic for accuracy [45]. Multiple messaging applications, such as WhatsApp, Telegram and WeChat, were found to be a key channel for the dissemination of health misinformation [46,47,48,49]. Moreover, a cross-platform comparison shows that misinformation on WhatsApp and Facebook has different forms (e.g., visual vs. links to extremist news sites) [50]. Thus, we shall treat messaging apps differently from other social media platforms.

#### 2.2.2. Reliance on Digital Health Media: Health Blogs and Alternative Health Media

Given that the coronavirus is essentially a health issue, people might also seek and obtain news and information from health-specific media such as health blogs and alternative health media, which could provide health-related information more intensely [17,51]. Blogs are defined as “web-based journals in which entries are published in reverse chronological sequence” [52]. Although sharing some features with social media such as commenting, blogs could provide longer-lasting content and are less textually limiting compared to, for example, Twitter [53]. Primarily aiming to provide information related to health and medical issues, health blogs may include disease-specific blogs covering a single topic and those that focus on the medical community in general [54].

Both credentialed and non-credentialed bloggers contribute to the health blogosphere. Researching 951 health blogs, Miller and Pole [55] found that nearly half of them were written by health professionals such as physicians and nurses, who provide less arcane but credible medical information for laypeople to care for themselves. On the other hand, patients are also actively involved in the blogosphere, covering a range of topics such as different types of chronic disease, and these blogs provide information based on their experience which could become an effective venue to gain health-related information [56]. However, there is no gatekeeping system in health blogs, so it is uncertain how accurate the information from these blogs is. Nevertheless, as an important source for health information, more exploration is needed to find out usage of health blogs under COVID-19 misinformation context.

Alternative health (*AH*) media, which was rarely investigated before, especially in the context of research on misinformation, might be a concern. *AH* media refers to “Web sites and blogs that provide information, news, and opinions as well as products related to alternative medicine” (Typical examples include traditional Chinese medicine, Ayurveda, homeopathy, and so forth) that are non-biomedical and non-standard treatments and rarely gain support from medical authorities [17]. During the past decades, alternative medicine has gained increasing popularity worldwide and some countries also resort to alternative medicine such as herbal medicine in the treatment of coronavirus [57]. COVID-19 pandemic and vaccines are also commonly discussed on *AH* media, such as “Natural News”, “Mercola.com” and “GreenMedInfo”, whereas the quality of online alternative medicine information was found to be generally low [58,59]. Moreover, content promoting alternative medicine is often linked to anti-vaccine websites [60,61]. Likewise, the anti-vaccine sentiment was often associated with using alternative or non-western medicine from research analyzing 480 anti-vaccine web pages [60]. The latest research shows a strong correlation between AH media consumption and misperception of vaccines [11,58]. Therefore, we suspect that during the pandemic, AH media would transmit COVID-19 vaccine information that lacks scientific support, leading to an increase in the misperception of the vaccines.

Distinguishing these different digital media platforms as sources of information about COVID-19, together with the above-mentioned mediating role of misinformation belief, we propose the following hypotheses:

**H1.** 
*More COVID-19 news exposure from social media (H1a), messaging applications (H1b), or AH media (H1c) will be associated with lower probability to get COVID-19 vaccines.*


**H2.** 
*More COVID-19 news exposure from social media (H2a), messaging applications (H2b), or AH media (H2c) will be associated with stronger belief in vaccine-related misinformation.*


**H3.** 
*Belief in vaccine-related misinformation will mediate the relationship between COVID-19 news exposure from digital media and vaccination behavior. Specifically, more COVID-19 news exposure from social media (H3a), messaging applications (H3b), or AH media (H3c) will increase misinformation belief, which will in turn decrease the likelihood of getting COVID-19 vaccines.*


We also propose a research question for health blogs due to relative lack of research in this type of digital media.

**RQ1.** 
*How would more COVID-19 news exposure from health blogs be associated with vaccine-related misinformation belief (RQ1a) and COVID-19 vaccination behavior (RQ1b)? And how misinformation belief would moderate the relationship between health blogs reliance and COVID-19 vaccination behavior?*


### 2.3. News Literacy Behavior: A Tool to Combat Online Misinformation?

Individuals’ susceptibility to misinformation gained from digital media could be different according to how much knowledge and skills they have in navigating news environments, which is so-called news literacy (NL) [62]. Vraga and her colleagues [25] defined NL as “knowledge of the personal and social processes by which news is produced, distributed, and consumed, and skills that allow users some control over these processes”. Another related concept is news literacy behaviors (NLB), the performance and applications of NL, which occur when people engage with news content critically and mindfully [25]. For example, being exposed to diverse news sources [63], verifying news content [64], and checking sources or links [65] are typical news literacy behavior. Although related, NLB is differentiated from NL as individuals may be familiar with news literacy ideas and have knowledge and skills around news content, systems, and practices but unable to apply them when consuming news [66,67]. Therefore, NLB as core outcomes of new literacy programs may better reflect how news literate individuals are and how critically and accurately they can analyze news.

Nevertheless, empirical research mainly studied NL and found inconsistent conclusions regarding the impact of NL on misinformed beliefs. Higher news media literacy was associated with larger news skepticism [63], less endorsement of conspiracy theories [68], stronger ability to distinguish between legitimate journalism and fabricated news [69]. People with high NL tended to be more discerning when using social media for news and information, and relied more on certain cues such as news outlets and headlines to evaluate the credibility of news while having less reliance on cues like comments and likes that reflect the popularity of social media posts [62]. However, Jones-Jang et al. [70] found no significant relationship between NL and identifying false news posts. Overall, prior research measures NL by looking into individuals’ knowledge of news media industries, message, audience and effect, but lacks an investigation of the NLB. We summarized extant empirical studies on literacy in Table 1 to show the existing measurement of literacy related factor, which shows a lack of research in NLB. Therefore, we would focus on the direct outcome of news literacy programs (i.e., news literacy behavior) and accordingly propose the following hypothesis.

**H4.** 
*The association between reliance on digital media (i.e., social media, messaging applications, health blogs, and AH media) for COVID-19 news and vaccine-related misinformation belief will be moderated by news literacy behavior such that the positive relationship between COVID-19 news exposure to social media, messaging applications or alternative health media and misinformation belief will be stronger for those with fewer news literacy behaviors.*


Taken together, we advance the following moderated mediation hypothesis:

**H5.** 
*The indirect association between digital media reliance and vaccination behavior via vaccine misinformation belief will be moderated by news literacy behavior. (The proposed direction is same as in H4).*


We are also interested in exploring the moderating role of news literacy behavior in association with health blogs, so we propose RQ2 for health blogs.

**RQ2.** 
*How would news literacy behavior moderate the reliance on health blogs for COVID-19 news exposure and misinformation belief (RQ2a)? And how would the indirect association between health blogs reliance and vaccination behavior via vaccine misinformation belief be moderated by news literacy behavior (RQ2b)?*


The theoretical framework of this study is presented in Figure 1.

## 3. Methods

### 3.1. Study Design

To examine the proposed hypotheses and answer the research questions, we conducted a cross-sectional survey in both U.S. and South Korea. We present our methods and results following the norms of conducting online survey [72].

### 3.2. Data Collection and Sample

We collected the non-probability based online survey data in the U.S. from June 15 to 30, 2021, and in South Korea from September 3 to 6, 2021 (This study’s data and code are shared at the Open Science Framework (https://osf.io/n5r4d/?view_only=dc94eae9112a44e189e030ba2d2c59cc, accessed on 31 December 2022). Data was collected in cooperation with two professional research firms, Dynata in the U.S. and Macromill Embrain in South Korea, who possess large online survey panels. These firms have large online panels with diversified survey respondents who are recruited through a number of different ways (e.g., partnership recruitment through company rewards programs) [73]. We designed the questionnaires and requested the firms to handle online formatting and distribution. American participants accessed the questionnaires on the Qualtrics’ platform while Korean participants did so on Embrain’s platform. The firms provided the raw data in the form of Microsoft Excel files, which were further handled and analyzed by the authors using R.

Overall, 1886 participants finished the survey (*N* = 1336 [the U.S.] (The final US sample was slightslightly smaller (N = 1150) because some participants did not answer all key questions in the survey); *N* = 550 [South Korea] (The reason for sample size discrepancy is due to funding complications. We have no reason to believe that the results would be biased due to this discrepancy. We replicated the results on a randomly selected U.S. subsample that is of equivalent sample size with the Korean data collection (Appendix A))).

### 3.3. Measures

***Digital media reliance for COVID-19 news*** was measured through asking participants “since the COVID-19 pandemic began, have you used the following platforms to get news and information related to COVID-19: (1) Social media (e.g., Facebook, Twitter); (2) Messaging applications (e.g., WhatsApp); (3) Health blogs (The option of health blogs was provided along with other digital media, so respondents can distinguish it from those social media accounts providing health suggestions); (4) Alternative medicine websites”. Response options ranged from 1 (Never) to 5 (All the time).

***Belief in vaccine-related misinformation*** was measured by asking participants to rate on a five-point scale ranging from “Not at all true” (coded = 1) to “Extremely true” (coded = 5) how true are the following statements: (1) Vaccines contain dangerous ingredients that can cause autism; (2) Coronavirus vaccines can cause infertility; (3) It is safer to catch the coronavirus and obtain natural immunity than getting the vaccine. The three items adapted from Islam et al. [74] and Mandavilli [75] were widely disseminated in both countries in the mid of 2020. We averaged these three items to create an index of belief in COVID-19 vaccine misinformation ([USA] = 0.90; [Korea] α = 0.73).

***News literacy behaviors.*** Participants were asked to rate on a five-point scale ranging from “Never” (coded = 1) to “Very frequently” (coded = 5) “When you are reading digital news articles on the Internet, how often do you: (1) Cross-check with mass media sources like news agencies, newspapers, television; (2) Cross-check the information with online verification sites like Politifact; (3) Check the sources cited/references in the story; (4) Examine the numerical and statistical information in detail; (5) If provided, read the updates and changes to the news story”. The items were adapted from Vraga et al. [25] focusing on actual behaviors that could reflect news literacy. Responses to the 5 items were averaged to create an index ([USA] = 0.93; [Korea] α = 0.86).

***Vaccination behavior*** was measured by asking respondents, “Did you get a COVID-19 vaccine” (All adults over 18 have been eligible for COVID-19 vaccines since August 2020 in South Korea and April 2020 in the U.S., which is earlier than our data collection, so the question is applicable for all respondents in our survey). Two response options were provided (Yes [coded = 1]; No [Coded = 0]).

***Control variables.*** Socio-demographic control variables included gender (1 = Male; 2 = Female), age (in years), education, income, and political orientation (1 = “Extremely conservative”, 9 = “Extremely liberal”). To make models comparable, race was not controlled across the two countries. We also included individuals’ interest in health-related news and information as a theoretical control variable (We run an additional analysis removing this theoretical control variable, and the results did not change) (e.g., food, diets, wellness, and medicines) was shown to influence health beliefs in previous research [17]. Moreover, we created a dummy variable named “country” that denotes which country observations were collected from (0 = the U.S., 1 = South Korea). This variable was used to control for country-level differences in the models that were run on combined dataset for both countries.

The questionnaire was first created in English, and then was translated into Korean by coders fluent in both English and Korean. The translated questionnaire was then reviewed and revised by two of the authors.

### 3.4. Analysis Strategy

We use PROCESS macro model 4 in R to examine the simple mediation model (H3), and model 7 was used to verify the rest of the research questions and hypotheses [76], including the main relationships (H1–2) and the moderating role of NLB (Given that NLB is hypothesized as a moderator which is theoretically not directly related to vaccine outcomes, we did not include it as a covariate in baseline models. Nevertheless, we also tested models by including NLB as a control variable and no notable changes were observed except that the significant relationship between health blogs and vaccination behavior in US data became insignificant) (H4–5). When performing these analyses, we bootstrapped our sample to 1000 resamples with 95% Cis (We also run the same analysis using 5000 and 10,000 bootstraps, and the results did not change (Appendix C and Appendix D)). In the examination of the moderated mediation model, PROCESS centered the mean to 0, producing *p* values for moderated mediation at the range for moderator variable ±1 SD from the mean. This analysis strategy was first used in the combined dataset of two countries and then employed at the country level. Additional analyses are reported in Appendix E and Appendix F.

## 4. Results

### 4.1. Sample Characteristics and Descriptive Results

In our sample, the American respondents were 51% of males and 49% of females with a median of 45 years of age. The South Korean respondents were 50.9% of males and 49.1% of females with a median of 45 years of age. Table 2 presents the descriptive statistics of the key variables.

### 4.2. Main Association Analysis

As shown in Table 3, misinformation belief is positively associated with messaging application reliance (b = 0.16, SE = 0.03, t = 5.91, *p* < 0.001), and AH media reliance (b = 0.33, SE = 0.03, t = 12.05, *p* < 0.001); however, no significant association was found for the relationship between misinformation belief and (a) news exposure on social media (b = 0.02, SE = 0.02, t = 5.91, *p* = 0.44) and (b) health blogs (b = −0.03, SE = 0.03, t = −1.27, *p* = 0.21). Hence, H2b and H2c were supported, whereas H2a was denied.

The results further indicated that neither reliance on social media (b = 0.01, SE = 0.06, t = 0.24, *p* = 0.81) nor that on AH media (b = 0.03, SE = 0.08, t = 0.38, *p* = 0.70) was significantly associated with vaccination behavior. Firthermore, health blogs reliance only showed marginally significant association with vaccination behavior (b = 0.14, SE = 0.07, t = 1.92, *p* = 0.06). More reliance on messaging applications (b = 0.26, SE = 0.07, t = 3.52, *p* < 0.001) predicted higher probability to take COVID-19 vaccines. Moreover, there was a negative strong association between misinformation belief and vaccination behavior (b = −0.67, SE = 0.07, t = −9.65, *p* < 0.001). Therefore, H1 was not supported.

### 4.3. Simple Mediation Analysis

The criteria for building up mediating relationships is whether the indirect relationship (ab) is significant or not, therefore we focus on the mediation index (ab) rather than individual path a or b [77]. Results of PROCESS macro model 4 showed that the indirect relationship of messaging applications reliance (ab = −0.11, BootSE = 0.02, 95% CI = [−0.15, −0.07]) or AH media reliance (ab = −0.22, BootSE = 0.03, 95% CI = [−0.29, −0.17]) on vaccination behavior through misinformation belief was significant (see Table 4). When the 95% confidence interval excludes zero, the relationship is significant, which was the case. However, belief in misinformation did not mediate the association between reliance on social media (ab = −0.01, BootSE = 0.02, 95% CI = [−0.04, 0.02]) or on health blogs (ab = 0.02, BootSE = 0.02, 95% CI = [−0.02, 0.06]) and vaccination behavior. Thus, H3b and H3c were supported, but H3a was denied

### 4.4. Moderated Mediation Analysis

We used PROCESS model 7 by bootstrapping our sample to 1000 resamples with 95% CIs to test the proposed moderated mediation model [78]. Prior to showing the moderated mediation relationship, PROCESS model first demonstrated a significant two-way interaction between social media reliance and NLB on misinformation belief (b = 0.08, SE = 0.02, t = 5.10, *p* < 0.001). Likewise, the other three types of digital media also had significant interaction with NLB (see Figure 2). As shown in Figure 2A,C, with more reliance on social media or health blogs, individuals with less NLB were less likely to believe vaccine-related misinformation, while conversely, those with more NLB were more vulnerable to misinformation. Figure 2B,D showed that a higher level of NLB had stronger positive relationship between reliance on messaging applications or on AH media and misinformation belief. For all types of media reliance, when individuals relied on these platforms at minimum levels, there was no difference in their misinformation belief regardless of their level of NLB. Thus, H4 was rejected.

In addition to the two-way interaction, results further demonstrated a significant moderated mediation model, in which the association between news exposure on four different types of digital media and vaccination behavior mediated through belief in misinformation was further moderated by NLB (Social media: Moderated Mediation Index = −0.05, BootSE = 0.01, 95% CI = [−0.08, −0.03]; Messaging applications: Moderated Mediation Index = −0.06, BootSE = 0.02, 95% CI = [−0.10, −0.03]; Health blogs: Moderated Mediation Index = −0.07, BootSE = 0.02, 95% CI = [−0.10, −0.04]; AH media: Moderated Mediation Index = −0.08, BootSE = 0.02, 95% CI = [−0.12, −0.04]).

Specifically, as shown in Table 5, the conditional indirect relationship of social media reliance on vaccination behavior via misinformation belief was positive when NLB was at a low level, whereas it turned negative for a high value of NLB. As for messaging applications reliance, the negative conditional indirect relationship for high values of NLB was stronger compared to moderate values, but it was non-significant for low values of NLB. The positive conditional indirect relationship of health blogs on vaccination behavior was stronger for low values of NLB compared to moderate values, but it became non-significant for high levels of NLB. A significant and negative indirect association between AH media reliance and vaccination behavior was present regardless of individuals’ NLB. However, this negative association became stronger for people with more NLB compared to those with fewer NLB. Thus, H5 was not supported.

### 4.5. Country-Level Differences

Similar to the combined datasets, we found that reliance on AH media predicted stronger misinformation belief that was still negatively associated with vaccination behavior in the U.S. and South Korea datasets. However, other digital media showed inconsistent patterns at the country level (Table 3). As for moderated mediation analysis, the relationship between reliance on messaging applications or AH media and vaccination behavior was mediated by misinformation belief (Table 4). After adding news literacy behavior to the model, we found that the indirect association between reliance on four types of digital media and vaccination behavior through misinformation belief was further moderated by NLB based on the significant moderated mediation indexes (Table 5). The pattern was similar to that in the combined dataset (Appendix B). However, we did not find any significantly moderated mediation in the South Korea dataset.

## 5. Discussion

Although extensive research has found being exposed to and believing in misinformation about COVID-19 vaccines is a challenge for vaccine acceptance [79,80], how the countervailing factors such as news literacy could complicate “the information exposure—belief in vaccine misinformation—vaccination” path is yet to be unpacked to understand the communication of scientific information about COVID-19. Our study revealed a significant moderated mediation model where the indirect effects of diverse digital media reliance on vaccination behaviors via vaccine-related misinformation belief were conditioned on NLB. Several intriguing findings merit further discussions.

First, aligning with our expectations, our findings about distinct digital media highlight news exposure through some platforms, particularly messaging applications and AH media, could fuel misinformed beliefs. However, different from our hypotheses, more reliance on social media did not predict stronger belief in vaccine-related misinformation or fewer vaccination behaviors. Despite being different from our expectations, this result about social media should come as no surprise given that some research has indicated that different social media platforms exert different levels of influence on COVID-19 misinformed belief and vaccine willingness. Users of Instagram, YouTube, Snapchat, and TikTok in the UK have a lower willingness to be vaccinated, whereas no significant association is found for Facebook and Twitter [43]. Due to platform affordances and features, Twitter usage is negatively associated with COVID-19 conspiracy endorsement, but using Facebook and YouTube is more likely to accumulate misinformed beliefs [81]. Therefore, when treating social media as a whole, it could cancel out individual platforms’ relationships with misbeliefs and result in non-significant results overall.

To answer our research question regarding health blogs, we found it did not have a significant relationship with misinformation belief and vaccination behavior. It is understandable given that patients or laypeople rather than just medical professionals can contribute to the blogosphere, and although laypeople may provide first-hand information about certain diseases such as cancer based on their experience [56], topics like COVID-19 vaccines are unfamiliar for them, thus related information on health blogs might be less accurate and contain misinformation as well.

While social media is increasingly being regulated to remove and downplay the reach of misinformation, messaging applications and AH media emerge as less regulated grounds for misinformation. Our finding on messaging applications reflects their concerning role in disseminating misinformation, which might be due to their affordance including encrypted technology that protects all text, voice, and media messages, peer-to-peer communication, and group architecture [82]. These features result in high interactivity index for messaging applications and make it more detrimental to hoodwinking people [47]. Similarly, our study that brings health-related digital media into misinformation literature showed consumption of AH media was linked to heightened misperception of vaccines, and this finding is the most consistent one across two countries in our research. It also confirms previous research that indicates a strong positive correlation between reliance on AH media and health-related misinformation such as vaccines and genetically modified foods in various contexts [11,17,61,83]. This again alerts us to pay closer attention to this type of media, especially during the global pandemic rife with misinformation as exposure to it might lead to serious consequences on health matters.

Second, our research indicates misperception of vaccines, to the extent that it is fueled by digital media exposure, acts as an important barrier to vaccination. Specifically, consistent with hypotheses, we found that more reliance on messaging applications or AH media for COVID-19 news was associated with a stronger belief in misinformation, which further led to a lower likelihood to get COVID-19 vaccines. However, this indirect relationship through mediation was largely different from the direct relationship of media reliance on vaccinations where messaging applications reliance predicted stronger vaccination behaviors and AH media reliance was not associated with vaccinations. This shows us that this particular mediation is a partial mediation, such that misinformation beliefs play a mediating role, but do not mediate the entire association between messaging application reliance and vaccine behavior. This mediation suggests it is important to integrate misinformed beliefs to understand the influence of news exposure to different media on health-related behaviors. Although we found that exposure to digital media that can rather easily contain misinformation could fuel vaccine misperceptions and eventually lower vaccinations, we should be alerted that more fact-related news does not necessarily increase vaccination rate as well. Using Twitter data, Lyu et al. [84] found that more fact-related news was associated with lower vaccination rate in the U.S., which might be because that more fact-related news about the vaccines might raise not only more discussions but more concerns. Further, understanding this mediation process using HBM is in line with prior findings that indicate perceived barriers to self-reported vaccinations involve a misunderstanding of vaccines [13,34], and perceived barriers constantly mediate media exposure and behaviors in different contexts [14].

Unexpectedly, different from our hypothesis, our study did not show a protective effect of NLB in the current context. That is to say that NLB moderated the indirect relationship of digital media reliance on vaccination behaviors. Specifically, among those with more NLBs, the indirect association between reliance on messaging applications, AH media or social media, and vaccination behaviors was more negative, whereas the positive indirect association between reliance on health blogs and vaccination behaviors was attenuated. This is because individuals with more NLBs could be more susceptible to vaccine misinformation when they relied on digital media more. This finding is contrary to previous studies which have suggested that NL is a useful tool to save people from the trap of misinformation [23,63,68]. Nevertheless, the moderated mediation relationships reveal the significance of NLB given that only reliance on messaging applications and AH media showed a significant indirect association with vaccination behaviors in the mediation results, whereas after taking NLB into consideration, we found that all types of digital media could have indirect effects on vaccinations through misinformed belief depending on the level of NLB. Hence, NLB is still shown to be an important factor.

Although counterintuitive, current findings for NLB might be explained by motivated reasoning [85,86]. Improving news literacy aims to limit reliance on simple, heuristic, or affective cues when individuals process information, increase the thoughtful and systematic routes in messaging processing, hone critical thinking ability, and become more cognitively sophisticated [87]. However, cognitive sophistication could magnify biased cognitive processing as people tend to be involved in motivated reasoning. So people with greater cognitive sophistication might bolster their views that align with their identities [88]. Therefore, more NLB might not help people uncover the truth but strengthen misperceptions if they had prior inclinations toward misinformed beliefs about vaccines. This points to the importance of incorporating the pre-existing attitudes towards vaccines into these models in future research to investigate the causes of this counterintuitive finding.

Another potential explanation for this result may be due to individuals’ overconfidence. NLB in our study was measured through self-reporting, but people’s feelings about their news literacy might be untethered from their actual experiences and behaviors [89]; thus it is possible that some people have a false sense of confidence when it comes to their levels of NLB. Those who have relatively high levels of self-perceived NLB might perceive they have sufficient behavior to evaluate news and information and therefore no further development of the tactics is needed [89]. Research also found that three in four Americans overestimate their ability to distinguish false news, which in turn influences their perceptions and behaviors such as visiting untrustworthy websites, sharing false information on social media, and believing those contents [90]. On the other hand, our results suggest that individuals with weaker NLB tend to be less susceptible to vaccine-related misinformation as they get more information on social media or health blogs. Ironically, because these people perceive themselves to be weak in terms of NLB, they may be extra cautious when processing information on these newer platforms, thus becoming more immune to misbeliefs. Simultaneously, this calls for future research using more diverse measurement strategies including cross-validation of behavioral news literacy items.

In terms of societal differences, we only observed the significant moderated mediation model in the U.S, not in the South Korea sample. In making sense of these results, we can first consider the different vaccination rates and sample sizes in Korea vs. the U.S. While data from both countries were collected after seven months since vaccination started in each country, Korea’s vaccination rate was higher than that of the U.S., marking 83% in September 2021 [91]. The vaccination rate in the U.S. in July 2020 was 67% [92]. The insignificant results in Korea may be in part due to potential ceiling effects given the high vaccination rates or smaller sample size. In addition, researchers found that scarcity in COVID-19 vaccine supplies may increase vaccine hesitancy [93], so the vaccine shortage faced by South Korea might influence the public’s perception towards the COVID-19 vaccine, leading to the different results between the U.S. and South Korea samples. Furthermore, public attitudes towards the COVID-19 vaccine have become highly politically polarized in the U.S. [94], which might also result in the difference in our study. Furthermore, this discrepancy might be attributed to cultural differences. Prior research found a stronger relationship between exposure to misinformation and information avoidance in the South Korea sample compared to the U.S. sample [95], which reflects the high-uncertainty avoidance culture in South Korea [96]. Therefore, we suspect that our Korean respondents might tend to avoid some COVID-19 misinformation given that popular misinformation usually contains more persuasive and uncertain words compared to accurate information [97]. News consumption habits could vary across different cultures, providing promising avenues for future research. Other possible explanations for the country difference might be vaccination campaigns implemented by different societies [93] or different technology adopted to promote vaccines [98], which opens more room for future research to explore.

As with all research, our study comes with a few limitations. First, the results are gained from cross-sectional surveys, so they only indicate a correlation rather than a causal relationship. To further examine these relationships, future research should consider using experimental data or panel survey data. Second, the self-reporting of variables such as media reliance for COVID-19 news and news literacy behavior might not reflect participants’ actual behaviors. People might not be able to accurately recall their time spent on getting COVID-19 related news from different media as it may be difficult to separate COVID-19 news from other types of information and from other aims of media usage like entertainment. Furthermore, as mentioned above, people may easily overestimate their levels of NLB. Although we have used an established NLB measure, we may consider adopting more objective measurements in future work. Third and relatedly, we only measured several behaviors and practices in NLB. We may consider other possible applications of news literacy in people’s daily lives such as checking news from unfamiliar sources, evaluating the reliability of news, questioning emotional reactions and bias, or using reverse image search, etc. [99]. In terms of measures of health blogs, we only asked participants about their usage of health blogs in general, whereas it might be possible that different types of health blogs may have various content quality. For example, blogs that focus on the medical information provided by health professionals could be more reliable than personal journal blogs that primarily are used to archive personal experiences and/or mental states related to personal matters. Therefore, future researchers may consider specifying the meaning of health blogs in questionnaires or examine different types of health blogs in detail. We may also include more items to measure vaccine misperception from multiple aspects. Last, future research could incorporate other factors related to vaccinations such as intentions to get vaccinated [100], and build a more well-rounded model to explain people’s health decisions.

## 6. Conclusions

This research shows a significant moderated mediation model, in which the association between online media use and COVID-19 vaccination was mediated through vaccine misperceptions, and this relationship was further moderated by news literacy behavior. Counterintuitively, among those with higher levels of news literacy behavior, the negative indirect association between reliance on social media, messaging applications, or alternative medicine websites and vaccinations was intensified. However, the positive indirect impact of health blogs was attenuated among individuals with stronger news literacy behaviors, compared to those with a low level of news literacy behaviors. Notably, we did not find the above significant relationship in the Korea dataset.

This study has two major practical implications. First, digital media varies in terms of how much it could contribute to people’s vaccine-related misperceptions, so determining which platforms should be paid more attention to is critical for decision-makers to efficiently allocate resources and formulate regulations. In our research, messaging applications and AH media showed a higher likelihood to mislead audiences, which alarms stakeholders that intense scrutiny should not only be on social media giants like Facebook and Twitter but also on other digital media with insidious possibilities to disseminate misinformation. For example, WhatsApp has attempted to limit the number of times a message can be simultaneously forwarded to other users [101].

Second, although the role of news literacy behaviors did not show a protective effect in the current study, it does not undermine the importance of news literacy education. What is worth reflecting on is what components should news literacy education include. Is educating students to check for different sources or reading the updates and changes to the news story sufficient for combating misinformation? This is in doubt especially when the online information environment might have already been contaminated and rife with misinformation, and people searching for more information from different sources could easily find other news containing similar perspectives due to digital echo chambers and feed algorithms. Although during the pandemic more people check public health websites for facts [99], we need to be cautious about it given that there lacks information rectification in the current online environment, and fact-checking tools are not popularized yet. Thus simply relying on external tools may not be enough to counter misinformation [102]; instead, efforts should be directed at building up internal and individualized fortresses such as critical thinking ability to undercut the influence of misinformation, or incorporating the training of discouraging motivated reasoning. Other practical literacy interventions may include teaching people more effective heuristics (e.g., skepticism toward catchy headlines), which should reduce reliance on low-effort processes [103]. In addition, obtaining news is not the only or the most important purpose to use media; instead, entertainment was selected as the number one reason for media use during the pandemic, especially for young people [99]. Thus, adopting various news literacy behavior could be time-consuming and tedious that might impede people from adopting news literacy behavior, so educators might think of adding gamification factors in their programs to better improve news literacy [104].

## Figures and Tables

**Figure 1 ijerph-20-00891-f001:**
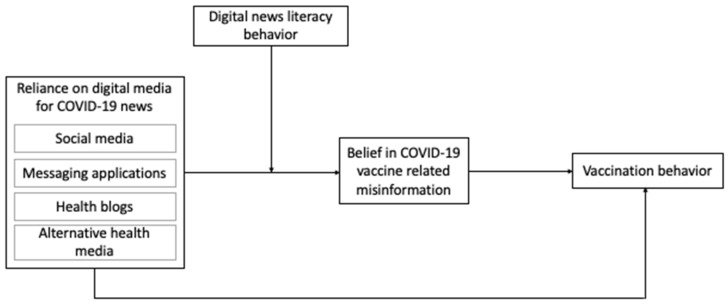
Theoretical framework.

**Figure 2 ijerph-20-00891-f002:**
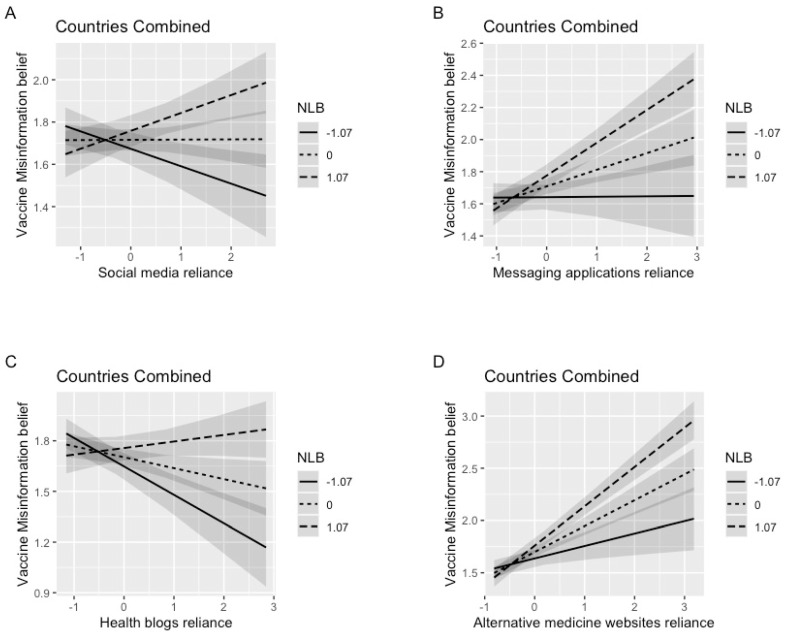
Two-way interaction between media reliance and NLB on vaccine-related misinformation belief (Combined sample). Note. Plots represent marginal effects at − 1 SD, Mean, and +1 SD. Figure **A**–**D** represents the interaction between NLB and reliance on (1) social media, (2) messaging applications, (3) health blogs and (4) alternative medicine website individually.

**Table 1 ijerph-20-00891-t001:** Summary of the extant empirical studies on literacy.

The Role of Literacy-Related Factor	Article	Sample	Measures of Literacy-Related Factor	Main Findings
Predictor	[22]	*N* = 546 [Dutch]; *N* = 545 [U.S.]	Media literacy intervention, 3 items (behavior ^1^)	Exposure to a media literacy intervention only has a significant effect on the perceived accuracy of misinformation, and not on issue agreement. A combination of a media literacy intervention and a fact-checker that refutes falsehoods is most effective.
	[70]	*N* = 1299 [U.S.]	Media literacy, 4 items (behavior) Information literacy, 5 items (knowledge) News literacy, 6 items (knowledge) Digital literacy, 10 items (skills)	The research showed that only information literacy—but not other literacies—significantly increased the likelihood of identifying fake news stories.
	[69]	*N* = 770 [U.S.]; N = 1067 [U.S.]	Procedural News Knowledge, 10 items (knowledge)	News expertise significantly influences how individuals evaluate online information, especially in political contexts. An understanding of how the news media operate can help individuals recognizing legitimate journalism from fabricated news and commercialized content.
	[68]	*N* = 397 [U.S.]	News media literacy, 16 items (knowledge & personal locus)	Individuals who have stronger belief in conspiracy theories know comparatively little about how the news media work. The greater one’s knowledge about the news media—from the kinds of news covered, to the commercial context in which news is produced, to the effects on public opinion news can have—the less likely one will believe in conspiracy theories.
	[63]	*N* = 508 [U.S.]	News media literacy, 26 items (knowledge & personal locus)	Highly news media literate teens had higher intrinsic motivation scores and would be more sceptical of the news media and more knowledgeable about current events than their less news media literate peers, but there was no difference in consuming any individual news medium between teens with high or low news media literacy.
Moderator	[24]	*N* = 571 [Unknown]	Critical consuming literacy, 10 items (capacity & skills)	The study revealed a path from incidental online news exposure to misinformation sharing, mediated by misperceptions. The mediated path was further moderated by narcissism and media literacy.
	[15]	*N* = 673 [U.S.]	News literacy, 9 items (knowledge)	Using social media to consume news content can influence vaccine hesitancy through increasing citizens’ scepticism regarding the efficacy of vaccines. However, these effects depended on the news literacy of users such that the effects on vaccine hesitancy are more substantial among those with lower news literacy.
	[71]	*N* = 875 [Taiwan]	Information literacy ^2^ (capacity & skills)	The research model showed that impact of adoption behavior of information and communication technology was moderated by information literacy. Specifically, low information literacy and high media experience affected ICT adoption behavior negatively, and low information literacy and high media technostress had a positive influence on ICT adoption behavior

Note. ^1^. We categorized the measures of literacy-related factors into different types (e.g., behavior, knowledge, skills) to represent from which perspective do these studies measure literacy. ^2^. The study does not provide the number of items to measure information literacy.

**Table 2 ijerph-20-00891-t002:** Descriptive Statistics.

Key Variables (Response Range)	Combined	US	South Korea
	M (SD)	M (SD)	M (SD)
Social media reliance (1–5)	2.30(1.36)	2.37(1.44)	2.15(1.16)
Messaging applications reliance (1–5)	2.06(1.34)	1.82(1.36)	2.59(1.15)
Health blogs reliance (1–5)	2.16(1.28)	2.12(1.36)	2.24(1.07)
AH media reliance (1–5)	1.81(1.20)	1.87(1.31)	1.69(0.91)
Misinformation belief (1–5)	1.78(1.12)	1.93(1.25)	1.46(0.67)
NLB (1–5)	2.69(1.07)	2.68(1.17)	2.70(0.82)
Vaccination behavior	66.74%	67.92%	64.18%

Note. Figures for vaccination behavior represent the percentage of people who got COVID-19 vaccines.

**Table 3 ijerph-20-00891-t003:** Regression models of the main direct effects.

	**Combined Dataset** (***N* = 1700**)						**U.S** (***N* = 1150**)		
	Model 1. Misinformation belief			Model 2. Vaccination behavior			Model 1. Misinformation belief		
	b	*p*	t	b	*p*	Z	b	*p*	t
constant	2.47	***	14.64	−2.59	***	−5.33	2.39	***	10.21
Social media	0.02		0.76	0.00		−0.06	0.02		0.67
Messaging applications	0.16	***	5.91	0.26	***	3.52	0.23	***	5.77
Health blogs	−0.03		−1.27	0.14	†	1.92	−0.05		−1.33
AH media	0.33	***	12.05	0.02		0.19	0.37	***	9.51
Age	−0.01	***	−5.78	0.04	***	8.36	−0.01	***	−3.42
Gender	−0.03		−0.58	−0.30	*	−2.41	−0.01		−0.12
Education	0.00		−0.08	0.21	***	4.79	−0.02		−0.96
Income	−0.01		−1.11	0.1	***	3.37	0.00		−0.25
Interest in health-related news	−0.01		−0.54	0.12	*	2.04	−0.07	*	−2.56
Political orientation	−0.14	***	−12.79	0.11	***	3.5	−0.14	***	−10.77
Misinformation belief				−0.67	***	−9.65			
Country	−0.77	***	−12.62	−0.28	†	−1.67			
** *Moderation* **									
Social media X NLB	0.08	***	5.1				0.08	***	4.12
Messaging applications X NLB	0.09	***	5.53				0.11	***	4.56
Health blogs X NLB	0.1	***	5.78				0.09	***	4.33
AH media X NLB	0.12	***	6.59				0.1	***	4.5
	**U.S** (***N* = 1150**)			**South Korea** (***N* = 550**)					
	Model 2. Vaccination behavior			Model 1. Misinformation belief			Model 2. Vaccination behavior		
	b	*p*	Z	b	*p*	t	b	*p*	Z
constant	−2.41	***	−3.74	1.94	***	9.33	−1.67	*	−2.07
Social media	−0.06		−0.73	0.02		0.77	0.02		0.2
Messaging applications	0.45	***	4.08	0.03		0.91	0.13		1.18
Health blogs	0.19	*	1.98	−0.04		−1.09	−0.02		−0.12
AH media	−0.13		−1.17	0.11	**	3.08	0.03		0.2
Age	0.03	***	4.66	0.00	*	−2.01	0.07	***	7.82
Gender	−0.22		−1.32	−0.03		−0.52	−0.52	*	−2.57
Education	0.26	***	4.52	0.01		0.55	0.13	†	1.81
Income	0.11	**	2.95	−0.05	**	−2.74	0.08		1.24
Interest in health-related news	0.15	*	2.02	0.09	**	3.14	−0.02		−0.22
Political orientation	0.12	**	3.3	−0.15	***	−5.81	−0.09		−0.95
Misinformation belief	−0.79	***	−9.25				−0.46	**	−2.94
Country									
** *Moderation* **									
Social media X NLB				−0.04		−1.59			
Messaging applications X NLB				−0.03		−0.97			
Health blogs X NLB				0.03		1.04			
AH media X NLB				0.05		1.52			

Note. Each column is a regression model that predicts the criterion at the top of the column. The baseline model results are from the regression analyses that do not include any interaction term. The interaction terms were tested separately in different regression analyses. Higher scores of political orientation represent being more liberal. Vaccination behavior was dummy coded such that 0 = did not get COVID-19 vaccine and 1 = got COVID-19 vaccine. Country was dummy coded such that 0 = the U.S. and 1 = South Korea, and this variable was only used in the combined dataset. † *p* < 0.10, * *p* < 0.05, ** *p* < 0.01, *** *p* < 0.001.

**Table 4 ijerph-20-00891-t004:** The mediation effect of media reliance for COVID-19 news on vaccination behavior.

	Combined Dataset (N = 1700)				U.S. (N = 1150)				South Korea (N = 550)			
	Effect	SE	LLCI	ULCI	Effect	SE	LLCI	ULCI	Effect	SE	LLCI	ULCI
Social media	−0.01	0.02	−0.04	0.02	−0.02	0.02	−0.07	0.03	−0.01	0.01	−0.04	0.01
Messaging applications	**−0.11**	**0.02**	**−0.15**	**−0.07**	**−0.18**	**0.04**	**−0.28**	**−0.11**	−0.01	0.02	−0.05	0.02
Health blogs	0.02	0.02	−0.02	0.06	0.04	0.03	−0.02	0.10	0.02	0.02	−0.02	0.06
Alternative health media	**−0.22**	**0.03**	**−0.29**	**−0.17**	**−0.29**	**0.05**	**−0.41**	**−0.20**	**−0.05**	**0.03**	**−0.11**	**−0.01**

Note. LLCI: lower limit confidence interval; ULCI: upper limit confidence interval. Significant effects in bold text.

**Table 5 ijerph-20-00891-t005:** The moderated mediation effect of media reliance for COVID-19 news on vaccination behavior.

	Combined dataset (N = 1700)					U.S (N = 1150)				Korea (N = 550)			
	Moderator value	Effect	BootSE	BootLLCI	BootULCI	Effect	BootSE	BootLLCI	BootULCI	Effect	BootSE	BootLLCI	BootULCI
Social media	Low NLB (−1 SD)	**0.06**	**0.02**	**0.01**	**0.10**	0.07	0.04	−0.00	0.14	−0.03	0.02	−0.08	0.01
	Moderate NLB (Mean)	0.00	0.02	−0.03	0.03	−0.01	0.03	−0.06	0.04	−0.01	0.01	−0.04	0.01
	High NLB (+1 SD)	**−0.06**	**0.02**	**−0.10**	**−0.02**	**−0.08**	**0.03**	**−0.15**	**−0.02**	0.00	0.02	−0.03	0.04
Messaging applications	Low NLB (−1 SD)	0.00	0.03	−0.06	0.05	0.01	0.06	−0.12	0.14	−0.02	0.02	−0.07	0.01
	Moderate NLB (Mean)	**−0.07**	**0.02**	**−0.11**	**−0.03**	**−0.09**	**0.04**	**−0.19**	**−0.01**	−0.01	0.02	−0.04	0.02
	High NLB (+1 SD)	**−0.14**	**0.03**	**−0.19**	**−0.09**	**−0.20**	**0.04**	**−0.29**	**−0.12**	0.00	0.03	−0.05	0.06
Health Blogs	Low NLB (−1 SD)	**0.11**	**0.03**	**0.06**	**0.17**	**0.15**	**0.05**	**0.07**	**0.27**	0.03	0.02	−0.01	0.08
	Moderate NLB (Mean)	**0.04**	**0.02**	**0.01**	**0.08**	**0.07**	**0.03**	**0.01**	**0.14**	0.02	0.02	−0.02	0.06
	High NLB (+1 SD)	−0.03	0.02	−0.08	0.02	−0.01	0.03	−0.08	0.05	0.01	0.03	−0.04	0.07
AH media	Low NLB (−1 SD)	**−0.08**	**0.04**	**−0.16**	**−0.01**	**−0.14**	**0.06**	**−0.26**	**−0.02**	−0.02	0.03	−0.09	0.03
	Moderate NLB (Mean)	**−0.17**	**0.03**	**−0.23**	**−0.11**	**−0.23**	**0.05**	**−0.34**	**−0.15**	−0.04	0.02	−0.09	0.00
	High NLB (+1 SD)	**−0.25**	**0.04**	**−0.33**	**−0.19**	**−0.33**	**0.06**	**−0.46**	**−0.23**	−0.06	0.03	−0.13	0.00
Index of moderated mediation		Index	BootSE	BootLLCI	BootULCI	Index	BootSE	BootLLCI	BootULCI	Index	BootSE	BootLLCI	BootULCI
Social media		**−0.05**	**0.01**	**−0.08**	**−0.03**	**−0.06**	**0.02**	**−0.10**	**−0.03**	0.02	0.02	−0.01	0.06
Messaging applications		**−0.06**	**0.02**	**−0.10**	**−0.03**	**−0.09**	**0.03**	**−0.14**	**−0.04**	0.01	0.02	−0.02	0.06
Health Blogs		**−0.07**	**0.02**	**−0.10**	**−0.04**	**−0.07**	**0.02**	**−0.12**	**−0.03**	−0.01	0.02	−0.06	0.02
AH media		**−0.08**	**0.02**	**−0.12**	**−0.04**	**−0.08**	**0.03**	**−0.14**	**−0.03**	−0.02	0.03	−0.08	0.02

Note. PROCESS Model 7. LLCI: lower limit confidence interval; ULCI: upper limit confidence interval. NLB: news literacy behavior. Numbers between 0.000 and 0.004 are rounded off to 0.00. Numbers between −0.004 and 0.000 are rounded off to −0.00. Significant effects in bold text.

## Data Availability

The data are available from the corresponding author upon reasonable request.

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
