# Peer review of "COVID-19 News Exposure and Vaccinations: A Moderated Mediation of Digital News Literacy Behavior and Vaccine Misperceptions"

_ijerph, 2023, doi:10.3390/ijerph20010891_

Round 1

Reviewer 1 Report

I would like to congratulate the authors for the manuscript they present, but in my opinion could be improved. My comments are organized below for your consideration. I hope my comments are useful for the study author(s) and editorial staff.

#Title:

- The title seems too big which may not be appealing to the future reader, so I suggest you try to reduce it.

#1 Introduction:

- Line 38 – 39: “Even until now, there are still 32% of Americans not fully vaccinated yet (Ritchie et al., 2020)” – Could you explain, why have you cited: (Ritchie et al., 2020).

- Line 48 – 49: “Recent research showed that these platforms could fuel health misinformation and vaccine intentions as well (Stecula et al., 2022; Wu et al., 2022)”.

- Line 62 - 65: “During health crises such as the COVID-19 pandemic, accessing reliable health information becomes more challenging, whereas vaccine-related misinformation could be easily gained not only from social media but also from other digital platforms such as alternative health media (Stecula et al., 2020; Wu et al., 2022)”.

(Stecula et al., 2020; Stecula et al., 2022; Wu et al., 2022) - These are all your published articles, while I beleive that your expertise in this area is outstanding, in order to enrich the literature of your manuscript and to suport the diversity, I kindle advise you to cite more articles by other distinctive scholars in this field.

- The authors should explain why they put the paragraph between line 96-105 in the introduction. In my opinion, if they are results of their study they should be in the discussion.

#2 Literature Review:

- Line 125 - 126: “However, many studies only measured general media usage or reliance, rather than media reliance specifically for COVID-19 news (Wu et al., 2022)”.

- Line 180 – 181: “WhatsApp, the messaging applications popular in multiple countries, was found to be a key channel for the dissemination of health misinformation (Kuru et al., 2022)”.

- Line 218 – 221: “AH media refers to Web sites and blogs that provide information, news, and opinions as well as products related to alternative medicine that are non-biomedical and non-standard treatments and rarely gain support from medical establishments (Wu et al., 2022)”.

- Line 230 – 231: “The latest research shows a strong correlation between AH media consumption and misperception of vaccines (Stecula et al., 2022; Wu et al., 2022)”.

(Kuru et al., 2022; Stecula et al., 2022; Wu et al., 2022) - These are all your published articles, while I beleive that your expertise in this area is outstanding, in order to enrich the literature of your manuscript and to suport the diversity, I kindle advise you to cite more articles by other distinctive scholars in this field.

#3 Methods:

- The authors should create the sub-heading "3.1. Study design" in which they should indicate the type of study conducted. They also should mention if they used any guidelines to write the present manuscript according to the type of study.

- The authors should clarify how the data collection took place. So, please provide information in your manuscript about the way you access the participants, how the participants access the questionnaires, etc.

- The authors must move the information about the sample characterization (lines 325-328) for the beginning of the heading "4. Results", and they could create the sub-heading "4.1. Sample characteristics".

- The authors must move the statement " The descriptive summaries of the key variables are provided in Table 2." (lines 335-336) and also the Table 2 to the heading "4. Results"

#4 Results

- The Table 3 and the Table 5 don't appear completely in the manuscript, some information about South Korea is not visible.

- Line 516 – 518: “It also confirms previous research that indicates a strong positive correlation between reliance on AH media and health-related misinformation such as vaccines and genetically modified foods in various contexts (Stecula et al., 2022; Wu et al., 2022)”. These are all your published articles, while I beleive that your expertise in this area is outstanding, in order to enrich the literature of your manuscript and to suport the diversity, I kindle advise you to cite more articles by other distinctive scholars in this field.

- The authors must revise the sentence " Another potential explanation for this result may be due to individuals’ overconfidence. news literacy behavior in our study was measured through self-reporting,(...)" (lines 564-565).

- The sub-heading "4.5. Discussions" must be transformed in the heading "5. Discussion".

#5 Conclusions and Implications

- The authors must move the limitations of their study (lines 640-665) from the heading "5. Conclusions" to the final of the heading "Discussion".

- The heading numbering of "5. Conclusions and Implications" must be changed for "6. Conclusions".

- The authors must add to the beginning of the heading "Conclusions" the major conclusions they achieve with this study, before they mention the implications of the present study.

# References

- The References must follow the norms of ACS, as required by the IJERPH.

- The authors must fill the "Institutional Review Board Statement" with the information required by the IJERPH (for example: number of the approval of the Ethical Committee, Helsinki Declaration, ...). If this approval by an  Ethical Committee was not given, then this study is scientifically compromised and its publication becomes unfeasible, as it does not meet the required ethical principles.

- The authors must fill the "Informed Consent Statement" indicating that the participants filled an informed consent, as required by the IJERPH when applicable. Since the participants filled out questionnaires, they had to give their informed consent. If this consent was not given, then this study is scientifically compromised and its publication becomes unfeasible, as it does not meet the required ethical principles.

Author Response

We thank the editor and reviewers for giving us an opportunity to revise our paper. We are deeply grateful for your insightful and constructive comments. We have taken advantage of your comments in carefully preparing this revision. Added or revised parts were marked using the “Track Changes” function. We have included detailed explanations (in blue color) in the individual responses to your comments (in black color).

Reviewer 2 Report

In this paper, after revising the existing scientific literature on the topic, the Authors sought to analyse the mediating role of belief in vaccine-related misinformation and the moderating role of news literacy behaviour in relation to COVID-19 information exposure and vaccination behaviours. The indirect impacts of various digital media reliance on vaccination behaviours through vaccine-related misinformation beliefs were conditioned on news literacy behaviour, according to this study’s significant moderated mediation model.

The paper is well-written, with very few comments regarding this sphere. Moreover, it technically sounds. The statistical analyses seem to be well-conducted with the usage of proper tests and R packages. I have some comments and suggestions for the authors that need to be addressed.

Introduction

This section seems adequate for the purposes of the study. Additionally:

Lines 33-35: I would rephrase it, maybe with this period “The wide-spread acceptance of the vaccines cannot be taken for granted, despite the COVID-19 vaccine being considered one of the most important scientific breakthroughs and health interventions to give herd immunity and end the pandemic.”

Line 36: define the meaning of “fully vaccinated”.

Lines 37-38: For more linearity and clarity, I would say that “Referring to the same period, other countries such as South Korea itself had low vaccinated rates […]”.

Lines 38-39: Reference is from 2020. How could this reference say something about the end of 2022?

As a general point, I suggest evaluating this article, the results appeared to be interesting: Lyu H, Zheng Z, Luo J. Misinformation versus Facts: Understanding the Influence of News regarding COVID-19 Vaccines on Vaccine Uptake. Health Data Sci. 2022;2022:9858292. Published 2022 Mar 12. doi:10.34133/2022/9858292

Literature Review

In general, this part is too discursive. I suggest revising this part by cutting some repetitions and unnecessary information. Moreover:

Line 180: “application” instead of “applications”.

Lines 200-201: Try to find a more recent reference other than “Buis and Carpenter, 2009”.

Line 210: “covering” instead of “coving” if I understood correctly.

Line 221: Also here, try to find a more recent reference than “Sagaram et al., 2002”.

Line 227-228: “Moreover, content promoting alternative medicine 227 is often linked to anti-vaccine websites”. Is this phrase supported by more recent articles?

Table 1. “[“ instead of “{“

Methods

Was this questionnaire validated? Which methods and analyses were used to assess the questionnaire’s reliability and validity?  The Authors stated that some questions were adapted from other surveys, but still they were translated into Korean. Please clarify this part. Moreover:

Line 324: “,” instead of “)”

Results

Was this questionnaire validated? Which methods and analyses were used to assess the questionnaire’s reliability and validity?  The Authors stated that some questions were adapted from other surveys, but still they were translated into Korean. Please clarify this part. Moreover:

Line 324: “,” instead of “)”

References must be made under the Guidelines for Authors: https://www.mdpi.com/journal/ijerph/instructions.

Finally, since they were administered questionnaires, you should disclose an informed consent statement, at least. Moreover, it should be explained why the institutional review board statement is not applicable.

Author Response

(The authors gave the same response as above.)

Reviewer 3 Report

1) Authors may wish to expand the introduction in terms of the disruptions that the pandemic constituted for societies which underlines the necessity of vaccination to provide a pathway out of the COVID-19 emergency (doi.org/10.3390/su14159699; doi.org/10.1016/j.ijsu.2020.04.018). Authors here should set the setting of how COVID-19 has led to an unprecedented surge of publications across all areas of knowledge (doi.org/10.3389/fpubh.2022.811885; doi.org/10.3346/jkms.2022.37.e296) and the subsequent deluge of information which most often dominated the public discourse.

2) Authors should provide explicit definitions of the terms misinformation or misperception of COVID-19 vaccines as well as what views may be constituted as either. This will support the decision to construct the “Misinformation beliefs index” using only three variables. 

3) Authors discuss the risks of misinformation and misperception of the COVID-19 vaccines through a theoretical framework that shows a straight relationship between vaccine behaviour and information consumption. The implicit assumption seems to be that vaccine-hesitant behaviour reflects misinformation internalised by the individual however this hypothesis may not adequately explain all possible practical or other factors (i.e., doctor’s advice, intention to vaccinate in the future) that may contribute to this decision. Please explain how these are considered in the study.

4) Authors test the association of digital media reliance with misinformation beliefs and vaccination behaviour which both result in a positive relationship. The authors then use the negative relationship between misinformation belief and vaccination behaviour to answer their research question “In our research, messaging applications and AH media showed a higher likelihood to mislead audiences”. However, the recorded positive association in both vaccination acceptance and misinformation beliefs when messaging applications are concerned seems counterintuitive which might point to the inability to determine whether digital reliance in a certain medium is directly linked to the source of misinformation. Please address this point.

Author Response

(The authors gave the same response as above.)

Round 2

Reviewer 1 Report

I would like to congratulate the authors on the changes they have made to the manuscript

Reviewer 3 Report

Authors have answered all of my concerns